# Collaboration-aware Graph Neural Network for Recommender Systems

**Yu Wang**
Vanderbilt University
yu.wang.1@vanderbilt.edu

**Yuying Zhao**
Vanderbilt University
yuying.zhao@vanderbilt.edu

**Yi Zhang**
Vanderbilt University
yi.zhang@vanderbilt.edu

**Tyler Derr**
Vanderbilt University
tyler.derr@vanderbilt.edu

## Abstract

Graph Neural Networks (GNNs) have been successfully adopted in recommendation systems by virtue of the message-passing that implicitly captures collaborative effect. Nevertheless, most of the existing message-passing mechanisms for recommendation are directly inherited from GNNs without scrutinizing whether the captured collaborative effect would benefit the prediction of user preferences. To quantify the benefit of the captured collaborative effect, we propose a recommendation-oriented topological metric, Common Interacted Ratio (CIR), which measures the level of interaction between a specific neighbor of a node with the rest of its neighbors. Then we propose a recommendation-tailored GNN, Collaboration-Aware Graph Convolutional Network (CAGCN), that goes beyond 1-WL test in distinguishing non-bipartite-subgraph-isomorphic graphs. Experiments on six benchmark datasets show that the best CAGCN variant outperforms the most representative GNN-based recommendation model, LightGCN, by nearly 10% in Recall@20 and also achieves more than 80% speedup. Our code is available at https://github.com/YuWVandy/CAGCN.

## 1 Introduction

Recommendation aims to alleviate information overload through helping users discover items of interest [1, 2]. Given historical user-item interactions, the key of recommendation systems is to leverage the Collaborative Effect [3–5] to predict how likely users will interact with items. A common paradigm for modeling collaborative effect is to first learn embeddings of users/items capable of recovering historical user-item interactions and then perform top-k recommendation based on the pairwise similarity between the learned user/item embeddings.

Since user-item interactions can be naturally represented as a bipartite graph, recent research has started to leverage GNNs to learn user/item embeddings for the recommendation [5–7]. Two pioneering works NGCF [5] and LightGCN [7] leverage graph convolutions to aggregate messages from local neighborhoods, which directly injects the collaborative signal into user/item embeddings. However, weighting messages based on node degrees as LightGCN cannot fully remove the influence of unreliable interactions. Even though NGCF leverages the affinity score to weigh neighbors, such affinity score is still calculated based on the dot-product between embeddings that are optimized by the unreliable interactions. Despite the fundamental importance of capturing beneficial collaborative signals, the related studies are still in their infancy. To fill this crucial gap, we aim to customize message-passing for recommendations and propose a recommendation-tailored GNN, namely Collaboration-Aware Graph Convolutional Network, that selectively passes neighborhood information based on their Common Interacted Ratio (CIR). Our contributions are:

Y. Wang et al., Collaboration-aware Graph Neural Network for Recommender Systems (Extended Abstract). Presented at the First Learning on Graphs Conference (LoG 2022), Virtual Event, December 9–12, 2022.

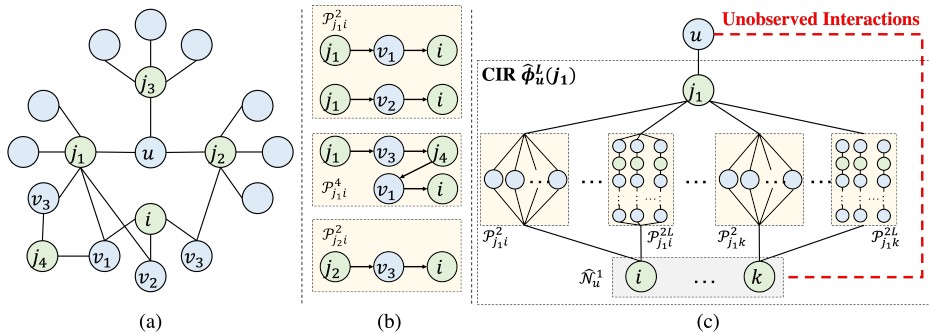

**Figure 1:** In (a)-(b), since $j_1, j_2$ have more interactions (paths) with (to) $i$'s neighbors than $j_3$, leveraging more collaborations from $j_1, j_2$ than $j_3$ would increase $u$'s ranking over $i$. In (c), we quantify the CIR between $j_1$ and $u$ via the paths (and associated nodes) between $j_1$ and $\widehat{\mathcal{N}}_u^1$.

- **Novel Recommendation-tailored Topological Metric:** We propose a recommendation-oriented topological metric, Common Interacted Ratio (CIR), and demonstrate the capability of CIR to quantify the benefits of aggregating messages from neighborhoods.
- **Novel Recommendation-tailored Graph Convolution:** We incorporate CIR into message-passing and propose a novel Collaboration-Aware Graph Convolutional Network (CAGCN). Then we prove that it can goes beyond 1-WL test in detecting non-bipartite-subgraph-isomorphic graphs, and demonstrate its superiority via comprehensive experiments on real-world datasets including two newly collected datasets and provide an in-depth interpretation on its advantages.

## 2 Method

In this section, we introduce notations used in this work, a novel recommendation-oriented topological metric (i.e., Common Interacted Ratio (CIR)) and then propose the collaboration-aware GNN.

**Preliminary.** Let $\mathcal{G} = (\mathcal{V}, \mathcal{E})$ be the user-item bipartite graph, where the node set $\mathcal{V} = \mathcal{U} \cup \mathcal{I}$ includes the user set $\mathcal{U}$ and the item set $\mathcal{I}$. User-item interactions are denoted as edges $\mathcal{E}$ where $e_{pq}$ represents the edge between node $p$ and $q$. The network topology is described by adjacency matrix $\mathbf{A} \in \{0, 1\}^{(|\mathcal{U}|+|\mathcal{I}|) \times (|\mathcal{U}|+|\mathcal{I}|)}$, where $\mathbf{A}_{pq} = 1$ when $e_{pq} \in \mathcal{E}$, and $\mathbf{A}_{pq} = 0$ otherwise. Let $\mathcal{N}_p^l$ and $\widehat{\mathcal{N}}_p^l$ denote the set of neighbors that are exactly $l$-hops away from $p$ in the training and testing set. Let $\mathcal{S}_p = (\mathcal{V}_{\mathcal{S}_p}, \mathcal{E}_{\mathcal{S}_p})$ be the neighborhood subgraph [8] induced in $\mathcal{G}$ by $\widetilde{\mathcal{N}}_p^1 = \mathcal{N}_p^1 \cup \{p\}$. We use $\mathscr{P}_{pq}^l$ to denote the set of shortest paths of length $l$ between node $p$ and $q$ and denote one of such paths as $P_{pq}^l$. Note that $\mathscr{P}_{pq}^l = \emptyset$ if it is impossible to have a path between $p$ and $q$ of length $l$, e.g., $\mathscr{P}_{11}^1 = \emptyset$ in an acyclic graph. Furthermore, we denote the initial embeddings of users/items in graph $\mathcal{G}$ as $\mathbf{E}^0 \in \mathbb{R}^{(n+m) \times d^0}$ where $\mathbf{e}_p^0 = \mathbf{E}_p^0$ is the node $p$'s embedding and let $d_p$ be the degree of node $p$.

### 2.1 Common Interacted Ratio

Graph-based methods capture collaboration from other users/items by message-passing. However, we cannot guarantee all of these collaborations benefit the prediction of users' preferences. For example, in Figure 1(a)-(b), given a center user $u$, we expect to leverage more collaborations from $u$'s observed neighboring items that have higher level of interactions (e.g., $j_1, j_2$ rather than $j_3$) with items that would be interacted with $u$ (e.g., $i$). To mathematically quantify such level of interactions, we propose a graph topological metric, Common Interacted Ratio (CIR):

**Definition 2.1.** *Common Interacted Ratio (CIR): For an observed neighboring item $j \in \mathcal{N}_u^1$ of user $u$, the CIR of $j$ around $u$ considering nodes up to $(L+1)$-hops away from $u$, i.e., $\widehat{\phi}_u^L(j)$, is defined as the average interacting ratio of $j$ with all neighboring items of $u$ in $\widehat{\mathcal{N}}_u^1$ through paths of length less than or equal to $2L$:*

$$\widehat{\phi}_u^L(j) = \frac{1}{|\widehat{\mathcal{N}}_u^1|} \sum_{i \in \widehat{\mathcal{N}}_u^1} \sum_{l=1}^{L} \beta^{2l} \sum_{P_{ji}^{2l} \in \mathscr{P}_{ji}^{2l}} \frac{1}{f(\{\mathcal{N}_k^1 | k \in P_{ji}^{2l}\})}, \forall j \in \mathcal{N}_u^1, \forall u \in \mathcal{U}, \tag{1}$$

where $\{\mathcal{N}_k^1 | k \in P_{ji}^{2l}\}$ represents the set of the 1-hop neighborhood of node $k$ along the path $P_{ji}^{2l}$ from node $j$ to $i$ of length $2l$. $\beta$ quantifies the importance/contribution of paths of length $2l$ connecting $i, j$. $f$ is a normalization function to differentiate the importance of different paths in $\mathscr{P}_{ji}^{2l}$ and its value

depends on the neighborhood of each node on the path $P_{ji}^{2l}$. As shown in Figure 1(c), the CIR of $j_1$ centering around $u$, $\widehat{\phi}_u^L(j_1)$ is decided by paths of length between 2 to $2L$. By configuring different $L$ and $f$, $\sum_{P_{ji}^{2l} \in \mathscr{P}_{ji}^{2l}} \frac{1}{f(\{\mathcal{N}_k^1 | k \in P_{ji}^{2l}\})}$ could express many existing graph similarity metrics [9–13] and we thoroughly discuss them in Appendix A.2. Calculating $\widehat{\phi}_u^L(j)$ is unrealistic since we do not have access to the testing set $\widehat{\mathcal{N}}_u^1$ in advance. Thereby, we propose to approximate $\widehat{\phi}_u(j)$ by enumerating $i$ from the observed training set $\mathcal{N}_u^1$ instead of $\widehat{\mathcal{N}}_u^1$ and denote this estimated version as $\phi_u^L(j)$. Such approximation assumes that neighboring nodes interacting more with other neighboring nodes in the training set would also interact more with neighboring nodes in the testing set, which is verified in Appendix A.3. We further empirically rationalize that edges with higher $\phi_u(j)$ are more important to the recommendation performance in Appendix A.8.3.

## 2.2 Collaboration-Aware Graph Convolutional Network

In order to pass node messages based on the benefits of their corresponding collaborations, we develop Collaboration-Aware Graph Convolutional Network. The core idea is to strengthen/weaken the messages passed from neighbors with higher/lower estimated CIR to center nodes. To achieve this, we compute the edge weight as: $\mathbf{\Phi}_{ij} = \phi_i(j)$ when $\mathbf{A}_{ij} > 0$ (and 0 otherwise), where $\phi_i(j)$ is the estimated CIR of neighboring node $j$ centering around $i$. Note that unlike the symmetric graph convolution $\mathbf{D}^{-0.5}\mathbf{A}\mathbf{D}^{-0.5}$ used in LightGCN, here $\mathbf{\Phi}$ is asymmetrical: the interacting level of node $j$ with $i$'s neighborhood is likely to be different from the interacting level of node $i$ with $j$'s neighborhood. We further normalize $\mathbf{\Phi}$ and combine it with the LightGCN convolution:

$$\mathbf{e}_i^{l+1} = \sum_{j \in \mathcal{N}_i^1} g\left(\gamma_i \frac{\mathbf{\Phi}_{ij}}{\sum_{k \in \mathcal{N}_i^1} \mathbf{\Phi}_{ik}}, d_i^{-0.5} d_j^{-0.5}\right) \mathbf{e}_j^l, \forall i \in \mathcal{V} \tag{2}$$

where $\gamma_i$ is a coefficient that varies the total amount of message flowing to each node $i$ and controls the embedding magnitude of that node [14]. $g$ is a function combining the edge weights computed according to CIR and LightGCN. In Appendix A.4, we prove that under certain choice of $g$ and $\gamma_i$, CAGCN can go beyond 1-WL test in distinguishing non-bipartite-subgraph-isomorphic graphs. Following the principle of LightGCN that the designed graph convolution should be light and easy to train, all other components of our architecture except the message-passing is exactly the same as LightGCN, which is covered in Appendix A.1 and A.5.

# 3 Experiments

## 3.1 Experimental Settings

We used six datasets including two newly collected datasets from other domains. MF [15], NGCF [5], LightGCN [7], UltraGCN [6], and GTN [16] are baselines. More details about datasets, baselines and experimental setup are provided in Appendix A.7. Following [17, 18], we set the embedding dimension to be 64 and the negative sample to be 1 for our CAGCN to ensure a fair comparison. For the first model variant CAGCN, we set $g(A, B) = g(A)$ where we remove $B = d_i^{-0.5} d_j^{-0.5}$ to solely demonstrate the power of passing messages according to CIR and set $\gamma_i = \sum_{j \in \mathcal{N}_i^1} d_i^{-0.5} d_j^{-0.5}$ to ensure the same embedding magnitude. For the second model variant CAGCN*, we set $g$ as weighted sum and $\gamma_i = \gamma$ as a constant controlling the contributions of capturing different collaborations.

## 3.2 Experimental Results

Here we describe the main experimental result observations with detailed insights in Appendix A.8.

**Performance Comparison.** Performance of baselines are provided in Table 1. We first compare the performance of LightGCN and CAGCN-variants. Clearly, CAGCN-jc/sc/lhn achieves higher performance than LightGCN because we selectively propagate node embeddings by the proposed CIR metrics (JC, SC, LHN). However, CAGCN-cn mostly performs worse than LightGCN because nodes having more common neighbors with other nodes are more likely to have higher degrees and hence aggregate more false-positive neighbors' information during message-passing. Comparing CAGCN*-variants with other competing baselines, CAGCN*-jc/sc almost consistently achieves higher performance than other baselines except UltraGCN on Amazon. This is because UltraGCN allows multiple negative samples for each positive interaction. Since GTN [16] uses different embedding size, we exclusively compare our model and GTN in Table 5 in Appendix A.8.

**Efficiency Comparison.** As recommendation models will be eventually deployed in user-item data of real-world scale, it is crucial to compare the efficiency of the proposed CAGCN(*) with other

**Table 1:** Results on R@20 and N@20 (i.e., Recall and NDCG) with best and runner-up highlighted.

| Model | Metric | MF | NGCF | LightGCN | UltraGCN | CAGCN | | | | CAGCN* | | |
| --- | --- | --- | --- | --- | --- | --- | --- | --- | --- | --- | --- | --- |
| | | | | | | -jc | -sc | -cn | -lhn | -jc | -sc | -lhn |
| Gowalla | Recall@20 | 0.1554 | 0.1563 | 0.1817 | 0.1867 | 0.1825 | 0.1826 | 0.1632 | 0.1821 | 0.1878 | 0.1878 | 0.1857 |
| | NDCG@20 | 0.1301 | 0.1300 | 0.1570 | 0.1580 | 0.1575 | 0.1577 | 0.1381 | 0.1577 | 0.1591 | 0.1588 | 0.1563 |
| Yelp2018 | Recall@20 | 0.0539 | 0.0596 | 0.0659 | 0.0675 | 0.0674 | 0.0671 | 0.0661 | 0.0661 | 0.0708 | 0.0711 | 0.0676 |
| | NDCG@20 | 0.0460 | 0.0489 | 0.0554 | 0.0553 | 0.0564 | 0.0560 | 0.0546 | 0.0555 | 0.0586 | 0.0590 | 0.0554 |
| Amazon | Recall@20 | 0.0337 | 0.0336 | 0.0420 | 0.0682 | 0.0435 | 0.0435 | 0.0403 | 0.0422 | 0.0510 | 0.0506 | 0.0457 |
| | NDCG@20 | 0.0265 | 0.0262 | 0.0331 | 0.0553 | 0.0343 | 0.0342 | 0.0321 | 0.0333 | 0.0403 | 0.0400 | 0.0361 |
| Ml-1M | Recall@20 | 0.2604 | 0.2619 | 0.2752 | 0.2783 | 0.2780 | 0.2786 | 0.2730 | 0.2760 | 0.2822 | 0.2827 | 0.2799 |
| | NDCG@20 | 0.2697 | 0.2729 | 0.2820 | 0.2638 | 0.2871 | 0.2881 | 0.2818 | 0.2871 | 0.2775 | 0.2776 | 0.2745 |
| Loseit | Recall@20 | 0.0539 | 0.0574 | 0.0588 | 0.0621 | 0.0622 | 0.0625 | 0.0502 | 0.0592 | 0.0654 | 0.0658 | 0.0658 |
| | NDCG@20 | 0.0420 | 0.0442 | 0.0465 | 0.0446 | 0.0474 | 0.0470 | 0.0379 | 0.0461 | 0.0486 | 0.0484 | 0.0489 |
| WorldNews22 | Recall@20 | 0.1942 | 0.1994 | 0.2035 | 0.2034 | 0.2135 | 0.2132 | 0.1726 | 0.2084 | 0.2182 | 0.2172 | 0.2053 |
| | NDCG@20 | 0.1235 | 0.1291 | 0.1311 | 0.1301 | 0.1385 | 0.1384 | 0.1064 | 0.1327 | 0.1405 | 0.1414 | 0.1311 |
| **Avg. Rank** | Recall@20 | 9.83 | 9.17 | 7.33 | 4.17 | 4.67 | 4.33 | 8.83 | 6.17 | 1.67 | 1.50 | 3.33 |
| | NDCG@20 | 9.50 | 9.17 | 5.83 | 6.00 | 3.67 | 4.00 | 8.33 | 5.00 | 2.50 | 2.50 | 5.17 |

**CAGCN-jc** indicates CAGCN equipped with CIR calculated based on jc metric and more details are provided in Appendix A.2-A.7.4.

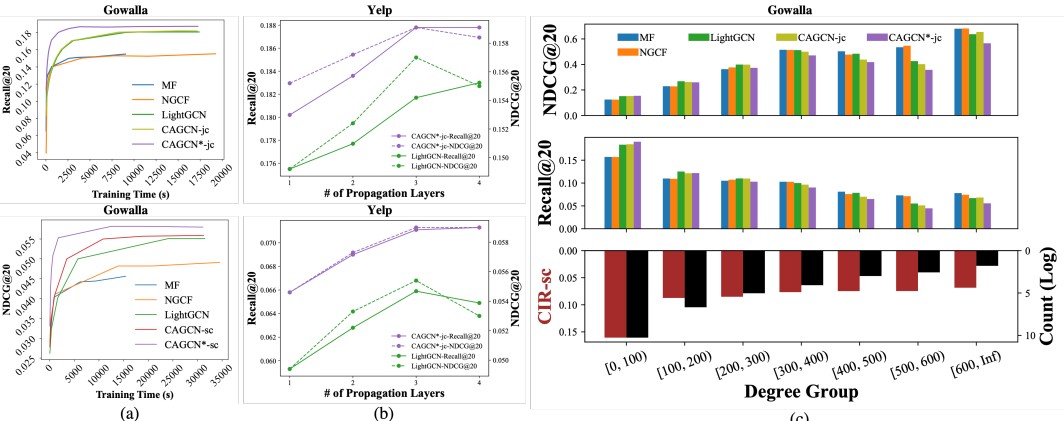

**Figure 2:** Efficiency comparison in column (a). Performance of different propagation layers in column (b). Performance of different models on users with different degrees in (c) where CIR-sc represents the CIR metric computed based on Salton Cosine Similarity and Count(Log) is the logarithm of the number of nodes.

baselines. For fair comparison, we use a uniform code framework implemented ourselves for all models and run them on the same machine. Clearly in Figure 2(a), CAGCN* achieves extremely higher performance in significant less time. This is because the designed graph convolution could recognize neighbors whose collaborations are most beneficial to users' ranking and by passing stronger messages from these neighbors.

**Impact of Propagation Layers.** We increase the propagation layer of CAGCN* and LightGCN from 1 to 4 and visualize their corresponding performance in Figure 2(b). The performance first increases as layer increases from 1 to 3 and then decreases on both datasets, which is consistent with findings in [7]. Our CAGCN* is always better than LightGCN at all layers.

**Interpretation on the advantages of CAGCN(*).** Here we visualize the performance of all models for nodes in different degree groups. Compared to non-graph-based methods (e.g., MF), graph-based methods (e.g., LightGCN, CAGCN(*)) achieve higher performance for lower degree nodes $[0, 300)$ while lower performance for higher degree nodes $[300, \text{Inf})$. Because the node degrees follow the power-law distribution [19], the average performance of graph-based methods would still be higher. On one hand, graph-based models could leverage neighborhood information to augment the weak supervision for low-degree nodes. On the other hand, it would introduce many noisy/unreliable interactions for higher-degree nodes. It is crucial to design an unbiased graph-based recommendation model that can achieve higher performance on both low and high degree nodes. In addition, the opposite performance trends between NDCG and Recall indicates that different evaluation metrics have different levels of sensitivity to node degrees.

## 4 Conclusion

In this paper, we propose the Common Interacted Ratio (CIR) to determine whether the captured collaborative effect would benefit the prediction of user preferences. Then we propose the Collaboration-

Aware Graph Convolutional Network to aggregate neighboring nodes' information based on their CIRs. We further define a new type of isomorphism, bipartite-subgraph-isomorphism, and prove that our CAGCN* can be more expressive than 1-WL in distinguishing subtree(subgraph)-isomorphic yet non-bipartite-subgraph-isomorphic graphs. Experimental results demonstrate the advantages of the proposed CAGCN(*) over other baselines. Specifically, CAGCN* outperforms the most representative graph-based recommendation model, LightGCN [7], by 9% in Recall@20 but also achieves more than 79% speedup. In the future, we plan to explore the imbalanced performance improvement among nodes in different degree groups as observed in Figure 2(c), especially from a GNN fairness perspective [20].

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

# A  Appendix

## A.1  Model Architecture and Training of LightGCN

Since our analysis is performed on the architecture of LightGCN, here we introduce the framework of LightGCN. Given the initial user and item embeddings $\mathbf{E}^0 \in \mathbb{R}^{(n+m) \times d^0}$, LightGCN performs L layers' message-passing as:

$$\mathbf{E}^l = \widetilde{\mathbf{A}}^l \mathbf{E}^0, \qquad \forall l \in \{1, 2, ..., L\}, \tag{3}$$

where $\widetilde{\mathbf{A}} = \widetilde{\mathbf{D}}^{-0.5} \mathbf{A} \widetilde{\mathbf{D}}^{-0.5}$ and $\widetilde{\mathbf{D}}$ is the degree matrix of $\mathbf{A}$. Then all $L$ layers propagated embeddings are aggregated together by mean-pooling:

$$\mathbf{E}^L = \frac{1}{(L+1)} \sum_{l=0}^{L} \mathbf{E}^l. \tag{4}$$

In the training stage, for each observed user-item interaction $(u, i)$, LightGCN randomly samples a negative item $i^-$ that $u$ has never interacted with before, and forms the triple $(u, i, i^-)$, which collectively forms the set of observed training triples $\mathcal{O}$. After that, the ranking scores of the user over these two items are computed as $y_{ui} = \mathbf{e}_u^\top \mathbf{e}_i$ and $y_{ui^-} = \mathbf{e}_u^\top \mathbf{e}_{i^-}$, which are finally used in optimizing the pairwise Bayesian Personalized Ranking (BPR) loss [15] and formalized as:$y_{ui} = \mathbf{e}_u^\top \mathbf{e}_i$ and $y_{ui^-} = \mathbf{e}_u^\top \mathbf{e}_{i^-}$

$$\mathcal{L}_{\text{BPR}} = \sum_{(u,i,i^-) \in \mathcal{O}} -\ln \sigma(y_{ui} - y_{ui^-}), \tag{5}$$

where $\sigma(\cdot)$ is the Sigmoid function, and here we omit the $L_2$ regularization term since it is mainly for alleviating overfitting and has no influence on collaborative effect captured by message passing.

## A.2  Graph Topological Metrics for CIR

Here we demonstrate that by configuring different $f$ and $L$, $\widehat{\phi}_u^L(j)$ can express many existing graph similarity metrics.

$$\widehat{\phi}_u^L(j) = \frac{1}{|\widehat{\mathcal{N}}_u^1|} \sum_{i \in \widehat{\mathcal{N}}_u^1} \sum_{l=1}^{L} \beta^{2l} \sum_{P_{ji}^{2l} \in \mathscr{P}_{ji}^{2l}} \frac{1}{f(\{\mathcal{N}_k^1 | k \in P_{ji}^{2l}\})}, \forall j \in \mathcal{N}_u^1, \forall u \in \mathcal{U}, \tag{6}$$

- **Jaccard Similarity (JC) [13]:** The JC score is a classic measure of similarity between two neighborhood sets, which is defined as the ratio of the intersection of two neighborhood sets to the union of these two sets:

$$\text{JC}(i, j) = \frac{|\mathcal{N}_i^1 \cap \mathcal{N}_j^1|}{|\mathcal{N}_i^1 \cup \mathcal{N}_j^1|} \tag{7}$$

Let $L = 1$ and set $f(\{\mathcal{N}_k^1 | k \in P_{ji}^{2l}\}) = |\mathcal{N}_i^1 \cup \mathcal{N}_j^1|$, then we have:

$$\widehat{\phi}_u^L(j) = \frac{1}{|\widehat{\mathcal{N}}_u^1|} \sum_{i \in \widehat{\mathcal{N}}_u^1} \beta^2 \sum_{P_{ji}^{2l} \in \mathscr{P}_{ji}^{2l}} \frac{1}{|\mathcal{N}_i^1 \cup \mathcal{N}_j^1|} = \frac{\beta^2}{|\widehat{\mathcal{N}}_u^1|} \sum_{i \in \widehat{\mathcal{N}}_u^1} \frac{|\mathcal{N}_i^1 \cap \mathcal{N}_j^1|}{|\mathcal{N}_i^1 \cup \mathcal{N}_j^1|} = \frac{\beta^2}{|\widehat{\mathcal{N}}_u^1|} \sum_{i \in \widehat{\mathcal{N}}_u^1} \text{JC}(i, j) \tag{8}$$

- **Salton Cosine Similarity (SC) [12]:** The SC score measures the cosine similarity between the neighborhood sets of two nodes:

$$\text{SC}(i,j) = \frac{|\mathcal{N}_i^1 \cap \mathcal{N}_j^1|}{\sqrt{|\mathcal{N}_i^1 \cup \mathcal{N}_j^1|}} \tag{9}$$

let $L = 1$ and set $f(\{\mathcal{N}_k^1 | k \in P_{ji}^{2l}\}) = \sqrt{|\mathcal{N}_i^1 \cup \mathcal{N}_j^1|}$, then we have:

$$\widehat{\phi}_u^L(j) = \frac{1}{|\widehat{\mathcal{N}}_u^1|} \sum_{i \in \widehat{\mathcal{N}}_u^1} \beta^2 \sum_{P_{ji}^{2l} \in \mathscr{P}_{ji}^{2l}} \frac{1}{\sqrt{|\mathcal{N}_i^1 \cup \mathcal{N}_j^1|}} = \frac{\beta^2}{|\widehat{\mathcal{N}}_u^1|} \sum_{i \in \widehat{\mathcal{N}}_u^1} \frac{|\mathcal{N}_i^1 \cap \mathcal{N}_j^1|}{\sqrt{|\mathcal{N}_i^1 \cup \mathcal{N}_j^1|}} = \frac{\beta^2}{|\widehat{\mathcal{N}}_u^1|} \sum_{i \in \widehat{\mathcal{N}}_u^1} \text{SC}(i,j) \tag{10}$$

- **Common Neighbors (CN) [11]:** The CN score measures the number of common neighbors of two nodes and is frequently used for measuring the proximity between two nodes:

$$\text{CN}(i,j) = |\mathcal{N}_i^1 \cap \mathcal{N}_j^1| \tag{11}$$

Let $L = 1$ and set $f(\{\mathcal{N}_k^1 | k \in P_{ji}^{2l}\}) = 1$, then we have:

$$\widehat{\phi}_u^L(j) = \frac{1}{|\widehat{\mathcal{N}}_u^1|} \sum_{i \in \widehat{\mathcal{N}}_u^1} \beta^2 \sum_{P_{ji}^{2l} \in \mathscr{P}_{ji}^{2l}} 1 = \frac{\beta^2}{|\widehat{\mathcal{N}}_u^1|} \sum_{i \in \widehat{\mathcal{N}}_u^1} |\mathcal{N}_i^1 \cap \mathcal{N}_j^1| = \frac{\beta^2}{|\widehat{\mathcal{N}}_u^1|} \sum_{i \in \widehat{\mathcal{N}}_u^1} \text{CN}(i,j) \tag{12}$$

Since CN does not contain any normalization to remove the bias of degree in quantifying proximity and hence performs worse than other metrics as demonstrated by our recommendation experiments in Table 1.

- **Leicht-Holme-Nerman (LHN) [9]:** LHN is very similar to SC. However, it removes the square root in the denominator and is more sensitive to the degree of node:

$$\text{LHN}(i,j) = \frac{|\mathcal{N}_i^1 \cap \mathcal{N}_j^1|}{|\mathcal{N}_i^1| \cdot |\mathcal{N}_j^1|} \tag{13}$$

Let $L = 1$ and set $f(\{\mathcal{N}_k^1 | k \in P_{ji}^{2l}\}) = |\mathcal{N}_i^1| \cdot |\mathcal{N}_j^1|$, then we have:

$$\widehat{\phi}_u^L(j) = \frac{1}{|\widehat{\mathcal{N}}_u^1|} \sum_{i \in \widehat{\mathcal{N}}_u^1} \beta^2 \sum_{P_{ji}^{2l} \in \mathscr{P}_{ji}^{2l}} \frac{1}{|\mathcal{N}_i^1| \cdot |\mathcal{N}_j^1|} = \frac{\beta^2}{|\widehat{\mathcal{N}}_u^1|} \sum_{i \in \widehat{\mathcal{N}}_u^1} \frac{|\mathcal{N}_i^1 \cap \mathcal{N}_j^1|}{|\mathcal{N}_i^1| \cdot |\mathcal{N}_j^1|} = \frac{\beta^2}{|\widehat{\mathcal{N}}_u^1|} \sum_{i \in \widehat{\mathcal{N}}_u^1} \text{LHN}(i,j) \tag{14}$$

- **Resource Allocation (RA) [10]:** RA is very similar to SC. However, it removes the square root in the denominator and is more sensitive to the degree of node:

$$\text{RA}(i,j) = \sum_{k \in \mathcal{N}_i^1 \cap \mathcal{N}_j^1} \frac{1}{|\mathcal{N}_k^1|} \tag{15}$$

Let $L = 1$ and set $f(\{\mathcal{N}_k^1 | k \in P_{ji}^{2l}\}) = \prod_{k \in P_{ji}^{2l}/\{i,j\}} |\mathcal{N}_k^1|$, then we have:

$$\widehat{\phi}_u^L(j) = \frac{1}{|\widehat{\mathcal{N}}_u^1|} \sum_{i \in \widehat{\mathcal{N}}_u^1} \beta^2 \sum_{P_{ji}^{2l} \in \mathscr{P}_{ji}^{2l}} \frac{1}{\prod_{k \in P_{ji}^{2l}/\{i,j\}} |\mathcal{N}_k^1|} = \frac{\beta^2}{|\widehat{\mathcal{N}}_u^1|} \sum_{i \in \widehat{\mathcal{N}}_u^1} \sum_{k \in \mathcal{N}_i^1 \cap \mathcal{N}_j^1} \frac{1}{|\mathcal{N}_k^1|} = \frac{\beta^2}{|\widehat{\mathcal{N}}_u^1|} \sum_{i \in \widehat{\mathcal{N}}_u^1} \text{RA}(i,j) \tag{16}$$

We further emphasize that our proposed CIR is a generalized version of these five existing metrics and can be delicately designed toward satisfying downstream tasks. We leave such exploration on the choice of $f$ as one potential future work.

## A.3 Approximation of CIR

Calculating $\widehat{\phi}_u(j)$ is unrealistic since we do not have access to the testing set $\widehat{\mathcal{N}}_u^1$ in advance. Thereby, we propose to approximate $\widehat{\phi}_u(j)$ by enumerating $i$ from the observed training set $\mathcal{N}_u^1$ instead of $\widehat{\mathcal{N}}_u^1$ and denote this estimated version as $\phi_u(j)$. Such approximation assumes that neighboring nodes interacting more with other neighboring nodes in the training set would also interact more with neighboring nodes in the testing set. We empirically verify such approximation by comparing the ranking consistency among CIRs calculated from training neighborhoods (i.e., $\phi_u(j)$), from testing neighborhoods ((i.e., $\widehat{\phi}_u(j)$)) and from full neighborhoods (we replace $\widehat{\mathcal{N}}_u^1$ with $\mathcal{N}_u^1 \cup \widehat{\mathcal{N}}_u^1$ in (6)). Here we respectively use four topological metrics (JC, SC, LHN, and CN) to define $f$ and rank the obtained three lists. Then, we measure the similarity of the ranked lists between Train-Test and between Train-Full by Rank-Biased Overlap (RBO) [21]. The averaged RBO values over all nodes $v \in \mathcal{V}$ on three datasets are shown in Table 2. We can clearly see that the RBO values on all these datasets using all topological metrics are beyond 0.5, which verifies our approximation. The RBO value between Train-Full is always higher than the one between Train-Test because most interactions are in the training set.

**Table 2:** Average Rank-Biased Overlap (RBO) of the ranked neighbor lists between training (i.e., $\mathcal{N}_u^1$) and testing/full (i.e., $\widehat{\mathcal{N}}_u^1$ and $\mathcal{N}_u^1 \cup \widehat{\mathcal{N}}_u^1$, respectively) dataset over all nodes $u \in \mathcal{U}$

| Metric | Gowalla | | Yelp | | Ml-1M | |
|---|---|---|---|---|---|---|
| | Train-Test | Train-Full | Train-Test | Train-Full | Train-Test | Train-Full |
| JC | 0.604±0.129 | 0.902±0.084 | 0.636±0.124 | 0.897±0.081 | 0.848±0.092 | 0.978±0.019 |
| SC | 0.611±0.127 | 0.896±0.084 | 0.657±0.124 | 0.900±0.077 | 0.876±0.077 | 0.983±0.015 |
| LHN | 0.598±0.121 | 0.974±0.036 | 0.578±0.100 | 0.976±0.029 | 0.845±0.082 | 0.987±0.009 |
| CN | 0.784±0.120 | 0.979±0.029 | 0.836±0.100 | 0.983±0.023 | 0.957±0.039 | 0.995±0.006 |

## A.4 Expressiveness of CAGCN

Here we thoroughly prove that when $g$ is set to be MLP, CAGCN can be more expressive than 1-WL. First, we review the concepts of subtree-isomorphism and subgraph-isomorphism.

**Definition A.1.** *Subtree-isomorphism [8]: $\mathcal{S}_u$ and $\mathcal{S}_i$ are subtree-isomorphic, denoted as $\mathcal{S}_u \cong_{subtree} \mathcal{S}_i$, if there exists a bijective mapping $h : \widetilde{\mathcal{N}}_u^1 \to \widetilde{\mathcal{N}}_i^1$ such that $h(u) = i$ and $\forall v \in \widetilde{\mathcal{N}}_u^1, h(v) = j, \mathbf{e}_v^l = \mathbf{e}_j^l$.*

**Definition A.2.** *Subgraph-isomorphism [8]: $\mathcal{S}_u$ and $\mathcal{S}_i$ are subgraph-isomorphic, denoted as $\mathcal{S}_u \cong_{subgraph} \mathcal{S}_i$, if there exists a bijective mapping $h : \widetilde{\mathcal{N}}_u^1 \to \widetilde{\mathcal{N}}_i^1$ such that $h(u) = i$ and $\forall v_1, v_2 \in \widetilde{\mathcal{N}}_u^1, e_{v_1 v_2} \in \mathcal{E}_{\mathcal{S}_u}$ $iff$ $e_{h(v_1)h(v_2)} \in \mathcal{E}_{\mathcal{S}_i}$ and $\mathbf{e}_{v_1}^l = \mathbf{e}_{h(v_1)}^l, \mathbf{e}_{v_2}^l = \mathbf{e}_{h(v_2)}^l$.*

Then we theoretically demonstrate the equivalence between the subtree-isomorphism and the subgraph-isomorphism in bipartite graphs:

**Theorem 1.** *In bipartite graphs, two subgraphs that are subtree-isomorphic if and only if they are subgraph-isomorphic.*

*Proof.* We prove this theorem in two directions. Firstly ($\Longrightarrow$), we prove that in a bipartite graph, two subgraphs that are subtree-isomorphic are also subgraph-isomorphic by contradiction. Assuming that there exists two subgraphs $\mathcal{S}_u, \mathcal{S}_i$ that are subtree-isomorphic yet not subgraph-isomorphic in a bipartite graph, i.e., $S_u \cong_{subtree} S_i, S_u \not\cong_{subgraph} S_i$. By definition of subtree-isomorphism, we trivially have $\mathbf{e}_v^l = \mathbf{e}_{h(v)}^l, \forall v \in \mathcal{V}_{\mathcal{S}_u}$. Then to guarantee $\mathcal{S}_u \not\cong_{subgraph} \mathcal{S}_i$ and also since edges are only allowed to connect $u$ and its neighbors $\mathcal{N}_u^1$ in the bipartite graph, there must exist at least an edge $e_{uv}$ between $u$ and one of its neighbors $v \in \mathcal{N}_u^1$ such that $e_{uv} \in \mathcal{E}_{\mathcal{S}_u}, e_{h(u)h(v)} \notin \mathcal{E}_{\mathcal{S}_i}$, which contradicts the assumption that $S_u \cong_{subtree} S_i$. Secondly ($\Longleftarrow$), we can prove that in a bipartite graph, two subgraphs that are subgraph-isomorphic are also subtree-isomorphic, which trivially holds since in any graph, subgraph-isomorphism leads to subtree-isomorphism [8]. □

Since 1-WL test can distinguish subtree-isomorphic graphs [8], the equivalence between these two isomorphisms indicates that in bipartite graphs, both of the subtree-isomorphic graphs and subgraph-isomorphic graphs can be distinguished by 1-WL test. Therefore, to go beyond 1-WL in bipartite

graphs, we propose a novel bipartite-subgraph-isomorphism in Definition A.3, which is even harder to be distinguished than the subgraph-isomorphism by 1-WL test:

**Definition A.3.** *Bipartite-subgraph-isomorphism: $\mathcal{S}_u$ and $\mathcal{S}_i$ are bipartite-subgraph-isomorphic, denoted as $\mathcal{S}_u \cong_{bi-subgraph} \mathcal{S}_i$, if there exists a bijective mapping $h : \widetilde{\mathcal{N}}_u^1 \cup \mathcal{N}_u^2 \to \widetilde{\mathcal{N}}_i^1 \cup \mathcal{N}_i^2$ such that $h(u) = i$ and $\forall v, v' \in \widetilde{\mathcal{N}}_u^1 \cup \mathcal{N}_u^2, e_{vv'} \in \mathcal{E} \iff e_{h(v)h(v')} \in \mathcal{E}$ and $\mathbf{e}_v^l = \mathbf{e}_{h(v)}^l, \mathbf{e}_{v'}^l = \mathbf{e}_{h(v')}^l$.*

With the bipartite-subgraph-isomorphism defined, we prove the injective property in the following:

**Lemma 1.** *If $g$ is MLP, then $g(\{(\gamma_i \widetilde{\boldsymbol{\Phi}}_{ij}, \mathbf{e}_j^l)|j \in \mathcal{N}_i^1\}, \{(d_i^{-0.5} d_j^{-0.5}, \mathbf{e}_j^l)|j \in \mathcal{N}_i^1\})$ is injective.*

*Proof.* If we assume that all node embeddings share the same discretization precision, then embeddings of all nodes in a graph can form a countable set $\mathcal{H}$. Similarly, for each edge in a graph, its CIR-based weight $\widetilde{\boldsymbol{\Phi}}_{ij}$ and degree-based weight $d_i^{-0.5} d_j^{-0.5}$ can also form two different countable sets $\mathcal{W}_1, \mathcal{W}_2$ with $|\mathcal{W}_1| = |\mathcal{W}_2|$. Then $\mathcal{P}_1 = \{\widetilde{\boldsymbol{\Phi}}_{ij} \mathbf{e}_i | \widetilde{\boldsymbol{\Phi}}_{ij} \in \mathcal{W}_1, \mathbf{e}_i \in \mathcal{H}\}, \mathcal{P}_2 = \{d_i^{-0.5} d_j^{-0.5} \mathbf{e}_i | d_i^{-0.5} d_j^{-0.5} \in \mathcal{W}_2, \mathbf{e}_i \in \mathcal{H}\}$ are also two countable sets. Let $P_1, P_2$ be two multisets containing elements from $\mathcal{P}_1$ and $\mathcal{P}_2$, respectively, and $|P_1| = |P_2|$. Then by Lemma 1 in [8], there exists a function $f$ such that $\pi(P_1, P_2) = \sum_{p_1 \in P_1, p_2 \in P_2} f(p_1, p_2)$ is unique for any distinct pair of multisets $(P_1, P_2)$. Since the MLP-based g is a universal approximator [22] and hence can learn the function $f$, we know that $g$ is injective. $\square$

**Theorem 2.** *Let M be a GNN with sufficient number of CAGC-based convolution layers defined by (2). If $g$ is MLP, then M is strictly more expressive than 1-WL in distinguishing subtree-isomorphic yet non-bipartite-subgraph-isomorphic graphs.*

*Proof.* We prove this theorem in two directions. Firstly ($\Longrightarrow$), following [8], we prove that the designed CAGCN here can distinguish any two graphs that are distinguishable by 1-WL by contradiction. Assume that there exist two graphs $\mathcal{G}_1$ and $\mathcal{G}_2$ which can be distinguished by 1-WL but cannot be distinguished by CAGCN. Further, suppose that 1-WL cannot distinguish these two graphs in the iterations from 0 to $L - 1$, but can distinguish them in the $L^{\text{th}}$ iteration. Then, there must exist two neighborhood subgraphs $S_u$ and $S_i$ whose neighboring nodes correspond to two different sets of node labels at the $L^{\text{th}}$ iteration, i.e., $\{\mathbf{e}_v^l | v \in \mathcal{N}_u^1\} \neq \{\mathbf{e}_j^l | j \in \mathcal{N}_i^1\}$. Since $g$ is injective by Lemma 1, for $S_u$ and $S_i$, $g$ would yield two different feature vectors at the $L^{\text{th}}$ iteration. This means that CAGCN can also distinguish $\mathcal{G}_1$ and $\mathcal{G}_2$, which contradicts the assumption.

Secondly ($\Longleftarrow$), we prove that there exist at least two graphs that can be distinguished by CAGCN but cannot be distinguished by 1-WL. Figure 3 presents two of such graphs $S_u, S_u'$, which are subgraph isomorphic but non-bipartite-subgraph-isomorphic. Assuming $u$ and $u'$ have exactly the same neighborhood feature vectors $\mathbf{e}$, then directly propagating according to 1-WL or even considering node degree as the edge weight as GCN [23] can still end up with the same propagated feature for $u$ and $u'$. However, if we leverage JC to calculate CIR as introduced in Appendix A.2, then we would end up with $\{(d_u d_{j_1})^{-0.5} \mathbf{e}, (d_u d_{j_2})^{-0.5} \mathbf{e}, (d_u d_{j_3})^{-0.5} \mathbf{e}\} \neq \{(d_{u'}^{-0.5} d_{j_1'}^{-0.5} + \widetilde{\boldsymbol{\Phi}}_{u' j_1'}) \mathbf{e}, (d_{u'}^{-0.5} d_{j_2'}^{-0.5} + \widetilde{\boldsymbol{\Phi}}_{u' j_2'}) \mathbf{e}, (d_{u'}^{-0.5} d_{j_3'}^{-0.5} + \widetilde{\boldsymbol{\Phi}}_{u' j_3'}) \mathbf{e}\}$. Since $g$ is injective by Lemma 1, CAGCN would yield two different embeddings for $u$ and $u'$. $\square$

Theorem 2 indicates that GNNs whose aggregation scheme is CAGC can distinguish non-bipartite-subgraph-isomorphic graphs that are indistinguishable by 1-WL.

## A.5 Model Architecture CAGCN

The model architecture of our CAGCN is shown in Figure 4. We take a specific example of computing the ranking of user $u$ over item $i$. We first calculate the estimated CIR of each neighbor with respect to the rest of the corresponding neighborhoods as (2) and then we iteratively propagate neighbors' embeddings with the awareness of the collaboration benefits by following the calculated CIR. Then we weighted combine the propagated embeddings at each layer to obtain the aggregated embedding for $u$ and $i$ as (3). After that, we calculate their ranking based on the dot-product similarity. The optimization of CAGCN is the same as LightGCN shown in (5).

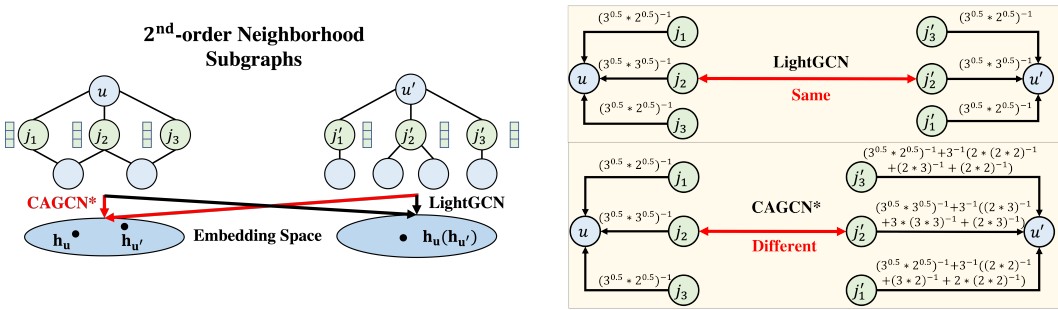

**Figure 3:** An example showing two neighborhood subgraphs $\mathcal{S}_u, \mathcal{S}_{u'}$ that are subgraph-isomorphic but not bipartite-subgraph-isomorphic.

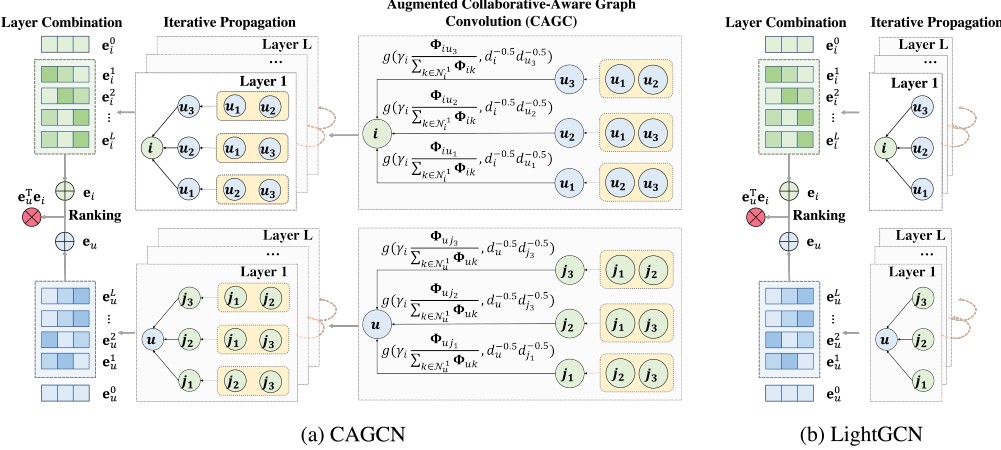

(a) CAGCN                    (b) LightGCN

**Figure 4:** Comparing the model architecture of CAGCN and LightGCN.

## A.6 Complexity Comparison and Analysis

Let $|\mathcal{V}|, |\mathcal{E}|, |\mathcal{F}|$ be the total number of nodes, edges, and feature dimensions (assuming feature dimensions stay the same across all feature transformation layers). Let $L$ be the propagation layer for all graph-based models using message-passing. Let $r$ be the total number of negative samples per epoch per positive pair and $K$ be the number of $2^{\text{nd}}$-order neighbors. For $r$, all baselines use 1 per epoch per positive pair and hence can be omitted (aside from UltraGCN using a larger number). Then the complexity of each model is summarized in Table 3. For CAGCN, since we only consider 2-hops away connections to compute CIR in Eq. (6), the main computational load would be computing the power of adjacency matrix, which takes $\mathcal{O}(|\mathcal{V}|^3)$. Note that for both of our CAGCN and UltraGCN, we can apply Strassens's Algorithm to further reduce the $\mathcal{O}(|\mathcal{V}|^3)$ to $\mathcal{O}(|\mathcal{V}|^{2.8})$. In Table 6, we report the preprocessing time for each dataset. Clearly, compared with the time used for training, the time for preprocessing is minor, which even demonstrates the superior efficiency of CAGCN since it significantly speeds up the training process.

**Table 3:** Complexity of the pre-procession and the forward pass of CAGCN and different baselines.

| Model | | MF | NGCF | LightGCN |
|---|---|---|---|---|
| # Extra Hyper-parameters | | / | / | 1 |
| Preprocess | Space | / | $\mathcal{O}(|\mathcal{E}| + |\mathcal{V}|)$ | $\mathcal{O}(|\mathcal{E}| + |\mathcal{V}|)$ |
| | Time | / | $\mathcal{O}(|\mathcal{E}| + |\mathcal{V}|)$ | $\mathcal{O}(|\mathcal{E}| + |\mathcal{V}|)$ |
| Training | Space | $\mathcal{O}(|\mathcal{V}|F)$ | $\mathcal{O}(L|\mathcal{V}|F + |\mathcal{E}| + LF^2)$ | $\mathcal{O}(L|\mathcal{V}|F + |\mathcal{E}|)$ |
| | Time | $\mathcal{O}(|\mathcal{E}|F)$ | $\mathcal{O}(L(|\mathcal{E}|F + |\mathcal{V}|F^2))$ | $\mathcal{O}(L|\mathcal{E}|F + L|\mathcal{V}|F)$ |

| Model | | GTN | UltraGCN | CAGCN |
|---|---|---|---|---|
| # Extra Hyper-parameters | | 1 | 7 | 2 |
| Preprocess | Space | $\mathcal{O}(|\mathcal{E}| + |\mathcal{V}|)$ | $\mathcal{O}(|\mathcal{E}| + |\mathcal{V}|)$ | $\mathcal{O}(|\mathcal{E}| + |\mathcal{V}|)$ |
| | Time | $\mathcal{O}(|\mathcal{E}| + |\mathcal{V}|)$ | $\mathcal{O}(|\mathcal{V}|^3)$ | $\mathcal{O}(|\mathcal{V}|^3)$ |
| Training | Space | $\mathcal{O}(L|\mathcal{V}|F + |\mathcal{E}|)$ | $\mathcal{O}(|\mathcal{V}|F + |\mathcal{V}|K)$ | $\mathcal{O}(L|\mathcal{V}|F + |\mathcal{E}|)$ |
| | Time | $\mathcal{O}(L|\mathcal{E}|F + L|\mathcal{V}|F)$ | $\mathcal{O}(r(|\mathcal{E}| + |V|K)F)$ | $\mathcal{O}(L|\mathcal{E}|F + L|\mathcal{V}|F)$ |

**Table 4:** Basic dataset statistics.

| Dataset | # Users | # Items | # Interactions | Density |
|---|---|---|---|---|
| Gowalla | 29, 858 | 40, 981 | 1, 027, 370 | 0.084% |
| Yelp | 31, 668 | 38, 048 | 1, 561, 406 | 0.130% |
| Amazon | 52, 643 | 91, 599 | 2, 984, 108 | 0.062% |
| Ml-1M | 6, 022 | 3, 043 | 895, 699 | 4.888% |
| Loseit | 5, 334 | 54, 595 | 230, 866 | 0.08% |
| WorldNews22 | 29, 785 | 21, 549 | 766, 874 | 0.119% |

## A.7  Experimental Setting

### A.7.1  Datasets

Following [5, 7], we validate the proposed approach on four widely used benchmark datasets in recommender systems, including **Gowalla**, **Yelp**, **Amazon**, and **Ml-1M**, the details of which are provided in [5, 7]. Moreover, we collect two extra datasets to further demonstrate the superiority of our proposed model in even broader user-item interaction domains: **(1) Loseit**: This dataset is collected from subreddit *loseit - Lose the Fat*[1] from March 2020 to March 2022 where users discuss healthy and sustainable methods of losing weight via posts. To ensure the quality of this dataset, we use the 10-core setting [24], i.e., retaining users and posts with at least ten interactions. **(2) WorldNews22**: This dataset includes the interactions from subreddit *World WorldNews*[2] where users share major WorldNews around the world via posts. Similarly, we use the 10-core setting to ensure the quality of this dataset. We summarize the statistics of all six datasets in Table 4.

### A.7.2  Baselines

We compare our proposed CAGCN with the following baselines:

- **MF [15]:** This is the most classic collaborative filtering method equipped with the BPR loss [15], which preserves users' ranking over interacted items with respect to uninteracted items.

- **NGCF [5]:** This was the very first GNN-based collaborative filtering model to incorporate high-order connectivity of user-item interactions for recommendation.

- **LightGCN [7]:** This is the most popular collaborative filtering model based on GNNs, which extends NGCF by removing feature transformation and nonlinear activation, and achieves better trade-off between the performance and efficiency.

- **UltraGCN [6]:** This model simplifies GCNs for collaborative filtering by omitting infinite layers of message passing for efficient recommendation, and it constructs the user-user graphs to leverage higher-order relationships. Thus, it achieves both better performance and shorter running time than LightGCN.

- **GTN [16]:** This model leverages a robust and adaptive propagation based on the trend of the aggregated messages to avoid the unreliable user-item interactions.

Note that here we only focus on baselines leveraging graph convolution (besides the classic MF) including the state-of-the-art GNN-based recommendation models (i.e., UltraGCN and GTN). There are some other developing methodology directions (e.g., [25–28]) that can obtain comparable results to the aforementioned baselines on some of the benchmark datasets. However, these methods are either not GNN-based [25] or incorporates some other general machine learning techniques rather than focus on graph convolution, e.g., SGCNs [28] leverages the stochastic binary masks to remove noisy edges, and GOTNet [27] performs k-Means clustering on nodes' embeddings to capture long-range dependencies. Given our main focus is on advancing the frontier of graph-convolution in recommendation systems, we omit these other comparable baselines. Note that our work could be further enhanced if incorporating these general techniques but we leave this as one future direction.

---

[1]https://www.reddit.com/r/loseit/
[2]https://www.reddit.com/r/worldWorldNews/

### A.7.3 Evaluation Metrics

Two popular metrics: Recall@K and Normalized Discounted Cumulative Gain (NDCG@K) [5] are adopted to evaluate all models. We set the default value of $K$ as 20 and report the average of Recall@20 and NDCG@20 over all users in the test set. In the inference phase, we treat items that the user has never interacted with in training set as candidate items. All recommendation models predict the user's preference scores over these candidate items and rank them based on the computed scores to further calculate Recall@20 and NDCG@20.

### A.7.4 Hyperparameter Settings

We strictly follow the experimental setting used in LightGCN [7] to ensure the fair comparison. For all other models, we adopt exactly the same hyper-parameters as suggested by the corresponding papers for all baselines to avoid any biased comparison: the embedding size $d^0 = 64$, learning rate $lr = 0.001$, the number of propagating layers $L = 3$, training batch size 2048. The coefficient of l2-regularization is searched in $\{1e^{-4}, 1e^{-3}\}$. As the user/item embedding is the main network parameter, it is crucial to ensure the same embedding size for fair comparison between different models. Therefore, when comparing with GTN [16], we set the embedding size to be 256 to align with [16]. For CAGCN, we set $\gamma_i$ as $\sum_{j \in \mathcal{N}_i^1} d_i^{-0.5} d_j^{-0.5}$ to ensure the same embedding magnitude as LightGCN. For $g$ and $\gamma_i$, CAGCN*, we set $g$ as the weighted sum in Eq. (2) for efficiency/less computation. Although using the weighted sum cannot guarantee the universal approximation of $g$ as MLP [22], we empirically find it still achieves superior performance over existing work. Furthermore, we set $\gamma_i = \gamma$ as a constant controlling the contributions of capturing different collaborations. Note that we search the optimal $\gamma$ within $\{1, 1.2, 1.5, 1.7, 2.0\}$. In addition, we term the model variant as CAGCN(*)-jc if we use JC to compute $\phi$.

### A.8 Additional Experimental Results

### A.8.1 Performance Comparison between CAGCN and GTN

Here we compare the performance between CAGCN and GTN. We first increase the embedding size $d^0$ to 256 following [16][3] and observe the consistent superiority of our model over GTN in Table 5. This is because in GTN [16], the edge weights for message-passing are still computed based on node embeddings that implicitly encode noisy collaborative signals from unreliable interactions. Conversely, our CAGCN* directly alleviates the propagation on unreliable interactions based on its CIR value, which removes noisy interactions from the source.

**Table 5:** Performance comparison of CAGCN* with GTN.

| Model | Metric | GTN | CAGCN* | | |
|---|---|---|---|---|---|
| | | | -jc | -sc | -lhn |
| Gowalla | R@20 | 0.1870 | 0.1901 | 0.1899 | 0.1885 |
| | N@20 | 0.1588 | 0.1604 | 0.1603 | 0.1576 |
| Yelp2018 | R@20 | 0.0679 | 0.0731 | 0.0729 | 0.0689 |
| | N@20 | 0.0554 | 0.0605 | 0.0601 | 0.0565 |
| Amazon | R@20 | 0.0450 | 0.0573 | 0.0575 | 0.0520 |
| | N@20 | 0.0346 | 0.0456 | 0.0458 | 0.0409 |

### A.8.2 Efficiency Comparison

As justified in Section 3, the efficiency plays a significant role in evaluating recommendation systems. As recommendation models will be eventually deployed in user-item data of real-world scale, it is crucial to compare the efficiency of the proposed CAGCN(*) with other baselines. For fair comparison, we use a uniform code framework implemented ourselves for all models and run them on the same machine with Ubuntu 20.04 system, AMD Ryzen 9 5900 12-Core Processor (2200 MHz), 128 GB RAM and GPU NVIDIA GeForce RTX 3090. Following the experimental setting in Figure 2(a), we present the NDCG@20 with the training time in Figure 5. Clearly, CAGCN*

---

[3]As the user/item embedding is the main network parameters, it is crucial to ensure the same embedding size when comparing different models and hence we use the exactly the same embedding size as GTN.

achieves extremely higher performance in significant less time because the collaboration-aware graph convolution leverages more beneficial collaborations from neighborhoods.

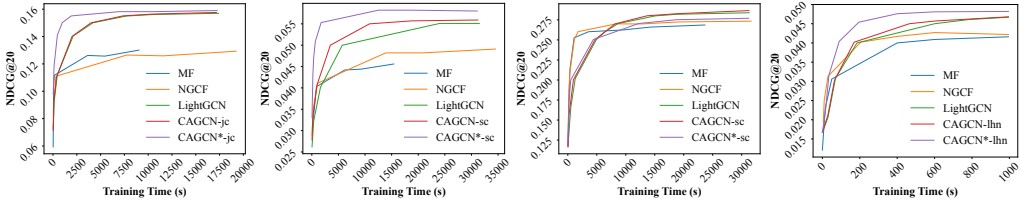

**(a)** Gowalla-NDCG@20    **(b)** Yelp2018-NDCG@20    **(c)** Ml-1M-NDCG@20    **(d)** Loseit-NDCG@20

**Figure 5:** Comparing the training time of CAGCN(*) with other baselines on four datasets. For clear visualization, we only report the efficiency of the best CAGCN(*) variant based on Table 1 for each dataset. CAGCN* almost always achieves extremely higher NDCG@20 with significant less time.

Furthermore, we report the first time that our best CAGCN* variant achieving the best performance of LightGCN on each dataset in Table 6. To ensure the fair comparison, we also include the time for precomputing CIR matrix as the preprocess time for our CAGCN*. We could see CAGCN* spends significant less time to achieve the same best performance as LightGCN, which highlights the broad prospects to deploy CAGCN* in real-world recommendations.

**Table 6:** Efficiency comparison of CAGCN* with LightGCN.

| Model | Stage | Gowalla | Yelp | Amazon | Ml-1M | Loseit | WorldNews22 |
|---|---|---|---|---|---|---|---|
| LightGCN | Training | 16432.0 | 28788.0 | 81976.5 | 18872.3 | 39031.0 | 13860.8 |
| CAGCN* | Preprocess | 167.4 | 281.6 | 1035.8 | 33.8 | 31.4 | 169.0 |
| | Training | 2963.2 | 1904.4 | 1983.9 | 11304.7 | 10417.7 | 1088.4 |
| | Total | 3130.6 | 2186.0 | 3019.7 | 11338.5 | 10449.1 | 1157.4 |
| **Improve** | Training | 82.0% | 93.4% | 97.6% | 40.1% | 73.3% | 92.1% |
| | Total | 80.9% | 92.4% | 96.3% | 39.9% | 73.2% | 91.6% |

### A.8.3 Empirical Analysis of CIR

To rationalize that edges with higher CIR would be more important to the recommendation performance. We leverage the LightGCN model with pre-trained user/item embeddings, remove all edges among nodes and add edges incrementally. Here we take two strategies: (1) **Global Strategy**: adding top-k edges among all edges in the whole graph according to their CIR; (2) **Local Strategy**: adding top-k edges among all edges around each node according to their CIR. Specifically for the local one, we first add the edges with highest CIR around each node and then add the edges with $2^{nd}$ highest CIR around each node and so on so forth. For both of these two strategies, we keep adding edges until the total number of added edges reach the predefined budget. We rank edges according to JC, SC, LHN and CO respectively and also compare them with randomly addition. We can clearly see that in most cases, adding edges with higher JC/SC/LHN would lead to better performance than random one, which demonstrates the importance of edges with higher JC/SC/LHN.

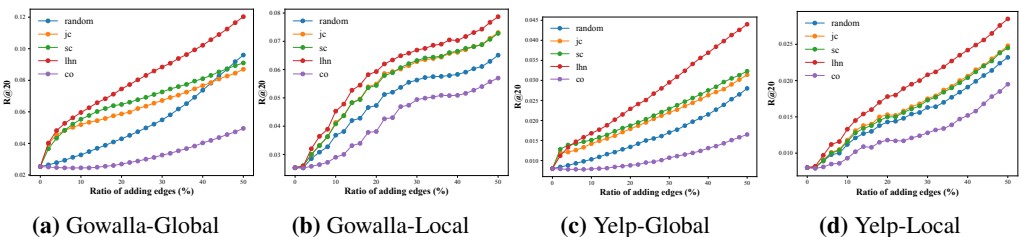

**(a)** Gowalla-Global    **(b)** Gowalla-Local    **(c)** Yelp-Global    **(d)** Yelp-Local

**Figure 6:** Performance of recommendation when adding edges randomly and according to different variants of CIR. (a) Adding edges

### A.9 Related Work

#### A.9.1 Collaborative Filtering (CF)

As an effective tool for personalized recommendation, CF assumes that people sharing similar interest on one thing tend to have the same preference on another thing, and it predicts the interest of a user (filtering) by utilizing the preference from other users who have similar interests (collaborative) [29]. Early CF methods used MF techniques [30], which generally map the IDs of users and items to a joint latent factor space and take the inner product of the embeddings to estimate the user-item interactions [15, 31]. Despite the initial success, these methods failed to capture the nonlinear user-item relationships due to their intrinsic linearity. To address this issue, deep learning was used to capture the non-linearity (e.g. by replacing the linear inner product operation with the nonlinear neural networks) [5, 32]. All above methods capture CF effect by optimizing embedding similarity based on observed user-item interactions. Stepping further, graph-based methods are proposed to leverage message-passing to directly inject the CF effect into the user/item embeddings [5, 7].

#### A.9.2 Graph-based Methods for Recommendation

Since user-item interaction can be naturally modeled as a bipartite graph, another line of research [5, 7, 33, 34] infers users' preferences by exploring the topological patterns of user-item bipartite graphs. Two pioneering work, ItemRank [33] and BiRank [34], define users' preferences based on their observed interacted items and perform label propagation to capture the CF effect. Although users' ranking scores are computed based on structural proximity between the observed items and the target item, the non-trainable user preferences and the lack of recommendation-based objectives in these methods lead to inferior performance to embedding-based methods such as MF-BPR [15]. Furthermore, HOP-Rec [35] combines the graph-based methods, which better capture the collaboration among nodes, and embedding-based methods, which better optimize the recommendation objective function. Yet, interactions captured by random walks do not fully explore the high-layer neighbors and multi-hop dependencies [36]. By contrast, GNN-based recommendation methods are superior at encoding structural proximity (especially higher-order connection) in user/item embeddings, which is crucial in capturing the CF effect [5, 7, 16]. For example, SGL [37] further leverages contrastive learning [38] with graph augmentation to enhance model robustness against noisy interactions, but it still follows the existing message-passing mechanism of GNNs without any justification. In fact, all of these GNN-based models directly borrow the traditional graph convolution operation from node/graph classification and blindly propagate neighboring users/items embeddings without any recommendation-tailored modification. Actually, our work has demonstrated that the collaboration captured by message-passing may not always improve users' ranking over items, which inspires us to design a new generation of graph convolutions that adaptively pass messages based on the benefits provided by the captured collaborations.

