# OpenReview forum: "Collaboration-Aware Graph Neural Network for Recommender Systems"
_logconference.io/LOG/2022/Conference — LoG 2022 Poster_

### Official Review · Reviewer_ZnaF · 2022-10-20

**Overall Score:** 6
**Confidence:** 5

**Review:**

### Summary

This paper presents Collaboration Aware Graph Convolutional Network aggregating neighboring nodes'' information based on their Common Interacted Ratio (CIR). The main contributions of this paper lie in (1) demonstrating the capability of CIR metrics to quantify the benefits of aggregating messages and (2) incorporating CIR into graph-based collaborative filtering models. The authors' experiments with six datasets demonstrated that the CAGCN variant outperforms the most representative recommendation model.

### Strength

1. The paper is well-organized and the notations are clearly written.
2. The paper includes comprehensive evaluation and improvements in terms of both accuracy and efficiency.
3. To explain the recommendation-oriented topological metric easily, this paper describes the example figures in detail.

### Weakness

1. The motivation for using the topological metric is weak. There is a lack of empirical results to confirm whether capturing harmful collaborative signals from unreliable interactions is the reason for damaging embeddings. Only the citation in line 30 shows that noisy interactions hinder performance.
2. There is insufficient support for the assumption in Line 75. For example, the assumption may be broken in a warm-start and a cold-start setting in a recommendation system. Therefore, clearer and more specific support are needed.
3. The baselines are insufficient. In the respective of degree-aware message-passing predictions, the authors need to compare with LinkProp[1]. LinkProp also mentions various existing degree-aware metrics (classic linkage scores) and shows differences from the metrics. However, in this paper, CIR uses the existing metrics LinkProp used. In addition, the author should describe the key differences from LinkProp or other degree-aware message-passing models.

> [1] Hao-Ming Fu, Patrick Poirson, Kwot Sin Lee, and Chen Wang. 2022. Revisiting Neighborhood-based Link Prediction for Collaborative Filtering. In Companion Proceedings of the Web Conference 2022 (WWW '22)

### Additional comments/feedback

As far as I understand, CIR in this paper uses existing topological metrics. Since applying CIR to the graph-based recommendation model alone does not contribute much, a rich analysis of each topological metric seems necessary. Although only one topological metric was shown for the degree group, it seems to provide a good insight if we compare and analyze the characteristics of each metric through a metric such as centrality.

### Recommendation

The reviewer favors designing a collaborative-aware recommendation model, a contribution that will be useful to the community. However, the authors did not thoroughly discuss the similar idea of the previous work. In addition, the authors need to add an analysis of several topological metrics in detail. Therefore, the reviewer cannot recommend acceptance of this paper in its current form.

---

### Official Review · Reviewer_HrxW · 2022-10-21

**Overall Score:** 8
**Confidence:** 3

**Review:**

To quantify the benefit of the captured collaborative effect, this paper proposes a recommendation oriented topological metric, Common Interacted Ratio (CIR), which measures the level of interaction between a specific neighbor of a node with the rest of its neighbors.  Then the authors propose a recommendation-tailored GNN, Collaboration-Aware Graph Convolutional Network (CAGCN), that goes beyond 1-WL test in distinguishing non-bipartite-subgraph-isomorphic graphs. Experiments on six benchmark datasets show that the best CAGCN variant outperforms the most representative GNN-based recommendation model, LightGCN. The proposed approach also achieves more than 80% speedup.

Although the proposed idea seems interesting, there are several concerns about the paper:
- (1) The authors should analysis the computational complexity of the proposed approach.
- (2) It seems that the proposed approach highly relies on data-preprocessing to obtain the statistics of paths and neighbors for CIR. The data-preprocessing seems to be extremely computational expensive, and the authors do not provide information about it. Therefore, it is unfair to say that the proposed approach also achieves more than 80% speedup without considering the data-preprocessing.
- (3) MF is a powerful baseline. However, some research [1][2] show that MF was not well implemented in some literature due to inappropriate experiment settings (such as the lack of L2 regularization on the vertex embeddings[1]). It seems that the authors adopt the performance under inappropriate experiment settings.

Therefore, my overall rating on this paper is weak reject.

- [1] Steffen Rendle, Walid Krichene, Li Zhang, and John Anderson. "Neural collaborative filtering vs. matrix factorization revisited." In Fourteenth ACM conference on recommender systems, pp. 240-248. 2020.
- [2] Desheng Cai, Jun Hu, Shengsheng Qian, Quan Fang, Quan Zhao, and Changsheng Xu. "GRecX: An Efficient and Unified Benchmark for GNN-based Recommendation." arXiv preprint arXiv:2111.10342 (2021).

---

### Official Review · Reviewer_8jdp · 2022-10-22

**Overall Score:** 6
**Confidence:** 4

**Review:**

This work proposes a metric named CIR to evaluate the importance of user-item interactions for collaborative filtering. Then author(s) propose to re-weight the user-item bipartite graph for effective and efficient recommendation.

**Pros:**
1. This work collects two new datasets for the community.
2. This paper conducts extensive experiments on six datasets to verify the effectiveness of the proposed method.
3. The author(s) give thorough analysis on the proposed metric CIR.

**Cons:**
1. The motivation is not clear. In line 28-29, the criticism towards existing GNN-based collaborative filtering methods seems not accurate. To me, it's not proper to criticize NGCF and LightGCN that they just follow the existing styles. They are also specifically designed for the recommendation task, but do not consider the properties that the author(s) mainly argued, i.e., CIR.
2. Presentation can be improved.
    1. \beta seems not defined in En. (1).
    2. The suffixes of CAGCN variants are not defined. The connection between variant suffixes and graph similarity metrics are not clearly
stated, making Table. 1 not self-containd.
    3. The legend of Figure. 2 (c) CIR-sc seems missed. What are the red and black bar charts mean?
    4. Over-claiming in line 132-133. There has been work showing the imbalance issues [1-2].

Overall, I vote for accepting this paper. Although the presentation could be improved, this paper studies an important question and proposes solid analysis and a promising approach. Besides, the author(s) collect two new datasets. The contribution is significant, especially in the extended-abstract track.

By the way, I personally suggest that the newly collected "News" dataset should have a more recognizable name/identifier. There have been too much datasets about news.

**Reference:**
> [1] Lin et al. Improving Graph Collaborative Filtering with Neighborhood-enriched Contrastive Learning. TheWebConf 2022.

> [2] Yu et al. Are Graph Augmentations Necessary? Simple Graph Contrastive Learning for Recommendation. SIGIR 2022.

---

### Official Review · Reviewer_Efhv · 2022-10-22

**Overall Score:** 8
**Confidence:** 5

**Review:**

This paper proposes to re-calculate the weights on edges to reflect higher-order collaborative signals in the user-item bipartite graph, which can improve the recommendation performance.

Strengths:

1 Reweighting the edge weights to reflect higher-order collaborative signals during training GNN-based CF method is a novel idea. There are also a lot of in-depth discussions on how to calculate the weights and what the impacts are when choosing different functions. The discussion is comprehensive and in detail.

2. The experiments are comprehensive and the improvements are convincing. The experiment are conducted on 6 benchmarks and the CAGCN is consistently better than other baselines, which justifies the effectiveness of considering CIR weights when learning GNN-based recommendation models

Weakness:

1. What is the time complexity when calculating the CIR score? It seems that the complexity of calculating this value is exponentially growing with the number of neighbors?

2 What is the meaning of $\beta$ in Eq (1) is not explained.

3 The author mentions the effectiveness of CAGCN* by weighted sum of both LightGCN weights and their CIR weights. What is the trends of performance when tuning the hyper-parameter?

Overall, this paper is well organized and of great technical contribution. Thus, I will vote to accept it.

---

### Meta-Review · Area_Chair_2gWF · 2022-11-18

**Confidence:** 4
**Recommendation:** Accept

**Meta Review:**

This work explores an interesting problem that handles higher-order collaborative relations by reweighting edges. The experiments and analyses are comprehensive. The authors have also done a good job in addressing the presentation and explanation issues during the rebuttal phase.

---

### Decision · Program_Chairs · 2022-11-22

Accept (Poster)